# Improving Turnaround Times for Routine Antimicrobial Sensitivity Testing Following European Committee on Antimicrobial Susceptibility Testing Methodology in Patients with Bacteraemia

**DOI:** 10.3390/antibiotics13111094

**Published:** 2024-11-17

**Authors:** Raewyn Edmondson, Kordo Saeed, Steve Green, Matthew O’Dwyer

**Affiliations:** 1Department of Infection, Microbiology, University Hospital Southampton, Tremona Road, Southampton SO16 6YD, UK; kordo.saeed@uhs.nhs.uk (K.S.); steve.green@uhs.nhs.uk (S.G.); matthew.o'dwyer@uhs.nhs.uk (M.O.); 2Clinical and Experimental Sciences, University of Southampton, Tremona Road, Southampton SO16 6YD, UK

**Keywords:** bacteraemia, empiric treatment, antimicrobial stewardship, antibiotic susceptibility testing, EUCAST

## Abstract

**Background/Objectives**: Bacteraemia can be fatal without antibiotic intervention. Antibiotic Susceptibility Testing (AST) provides the necessary information for targeted antibiotic therapy; however, the traditional method using disc diffusion can take over two days from a positive blood culture. Inappropriate empiric therapy is associated with increased mortality and increased antibiotic resistance, highlighting the need for more rapid turnaround times for AST. By making changes to an established method, turnaround times can be reduced. **Methods**: Eighty-two patient positive blood culture samples were collected from January to April 2022, representing the range of common bacteria causing sepsis. This followed the normal methodology in the laboratory of inoculating agar from positive blood cultures in preparation for European Committee on Antimicrobial Susceptibility Testing (EUCAST) disc diffusion AST method. EUCAST methodology outlines that disc diffusion should be performed on isolates from an overnight culture of 16–24 h. This study looked at comparing disc diffusion results from cultures with 6 h of incubation to those with incubation times of 24 h, after organism identification by MALDI-ToF. Results from 6-h and 24-h cultures were compared by disc zone sizes and by interpreted susceptibility reading following EUCAST guidelines of sensitive, resistant, susceptible with increased exposure, or an area of technical uncertainty. **Results**: A total of 99.65% interpreted susceptibility readings matched across all organisms to all relevant antibiotics, with an average zone size difference of 1.08 mm between results from 6 h versus 24 h cultures. **Conclusions**: This method offers a non-automated way of using the traditional disc diffusion method, reducing turnaround times while still producing reliable and accurate results. This would mean validated ASTs can be set up in the same day as a blood culture flags positive rather than waiting for a longer culture. As this method is widely used within the laboratory already, it would mean that additional training is not required, as the process is the same, and only incubation time varies. This would positively impact patient outlook due to the shorter use of empiric therapy, and benefit antimicrobial stewardship (AMS).

## 1. Introduction

Bacteraemia and associated sepsis are challenging clinical issues in healthcare settings and contribute to preventable deaths [1,2,3]. Bacteraemia may be community or healthcare-associated, with healthcare-associated being more commonly linked to organisms that display antibiotic resistance. *Escherichia coli* is the predominant Gram-negative cause of bacteraemia, while *Staphylococcus* spp. and *Streptococcus* spp. are the most common Gram-positive organisms [4,5,6]. These can be detected by fluorescence-based blood culture instrumentation such as the BD Bactec FX (Becton Dickinson, Franklin Lakes, NJ, USA), and once the infecting organism has been detected, identified, and AST is available, targeted treatment can commence.

The key treatment for bacterial bloodstream infection is antibiotic therapy, and without prompt intervention, septicaemia is highly likely [7,8]. Antimicrobial resistance (AMR) has been rapidly increasing since it was first observed. Resistance is likely to be exacerbated by the overuse or misuse of antibiotics, providing a selection pressure for resistant organisms [9].

In hospital laboratories, AST is performed to determine the best choice of treatment for bacteraemic patients while also contributing to AMS and limiting the use of unnecessary or ineffective antibiotics. When interpreting AST results, zones of inhibition are measured around an antibiotic disc on culture medium inoculated with the organism. Generally, NHS laboratories in the UK use standards set by EUCAST which are used to interpret zone measurements as resistant or susceptible for each organism and antibiotic [10].

The current process from blood culture bottle receipt in the laboratory to AST results has long turnaround times. From a blood culture flagging positive, the turnaround time can be up to three days, with the time to initial detection adding to this. Survival rates are highest the earlier appropriate antimicrobial therapy is administered [7,11]. Empirical treatment involves the early administration of broad range antibiotics targeting the most likely causative organism based on local epidemiology and the likely source of infection, while the causative organism is unknown. This aims to target the infectious agent before targeted antibiotic therapy can be administered following AST [12]. Empirical or broad-spectrum antibiotic therapy could promote AMR by employing the use of unnecessary antibiotics where the causative organism is wrongly predicted. Therefore, any intervention that can minimise turnaround times for AST would be considered beneficial for patient management and AMS.

EUCAST have a Rapid Antimicrobial Susceptibility Testing (RAST) method available, which can be prepared directly from the blood culture bottle, which can be read at 4, 6, or 8 h intervals [13]. This method was not selected for assessment as there were some limitations of this test that made in unsuitable for use in this laboratory. The method described by EUCAST only applied to the organisms *E. coli*, *Klebsiella pneumoniae*, *Pseudomonas aeruginosa*, *Acinetobacter baumannii*, *Staphylococcus aureus*, *Enterococcus faecalis*, *Enterococcus faecium*, and *Streptococcus pneumoniae*, and for each of these organisms, there is a limited number of antibiotics available which does not constitute the full panel tested at University Hospital Southampton (UHS). The incubation length varies isolate to isolate; at 4 h the success rate ranges from 44% to 99%; at 6 h, the success rate is 83–100%; and at 8 h, the success rate ranges from 95 to 100%. The disc diffusion inhibition zones also need to be read ±5 min of the incubation period of 4, 6, or 8 h, which is difficult to ensure in a busy tertiary hospital. The inconsistent incubation times and short windows to read the disc diffusion plates mean that this would be difficult to execute consistently and to plan workflows around. Even if the method was used and incubated for the maximum 16–20 h, there is still a limited number of organisms and selective antimicrobial combinations, and additional testing would be required to generate reportable data for the entire repertoire. The method listed in this manuscript was devised as an additional option to the RAST method for sites that require an extended repertoire of antimicrobial susceptibilities, meaning the entire repertoire of antibiotic sensitivities can be done without necessitating additional work 24 h later.

The aim of this study is to improve and reduce turnaround times for AST results in positive blood cultures, without applying expensive technology, at a tertiary hospital.

## 2. Results

Eighty-two patient samples were included in this study, as well as nine control samples. The bacterial isolates obtained from patient samples were 21 *Staphylococcus* spp. specimens, 20 Enterobacterales, 21 *Enterococcus/Streptococcus* spp., and 20 *Haemophilus influenzae/Moraxella catarrhalis* that underwent comparison of AST results from original 6 and 24 h cultures. All controls displayed expected resistance. Taking all antibiotics for each organism into account, there was a total of 858 comparisons.

The raw data are accessible through Appendix A. The following qualitative and quantitative results analysis is drawn from the raw data.

### 2.1. Qualitative Results

When comparing the 6 and 24 h culture AST results, a total of 855 out of 858 (99.65%) comparisons matched as either S, R, I, or Area of Technical Uncertainty (ATU) (Table 1). Statistical analysis of these results gave the following values: sensitivity 99.65%, specificity 99.68%, PPV 99.12%, NPV 99.84%, accuracy 99.65%.

### 2.2. Quantitative Results

When comparing antibiotic disc zone sizes in mm, the mean difference between the 6 and 24 h culture AST across all organisms for each antibiotic was 1.08 mm (standard deviation 1.33 mm). Figure 1 summarises these results for all organisms and antibiotics. Regression analysis shows that the correlation between zone sizes from 6 and 24 h incubation plates post-blood culture was significant (*p* < 0.05), while the small mean difference of 1.08 mm between zone sizes was not significant. Results were also not impacted by variables such as drug molecular weight, organism group, or sensitive/resistant strains.

## 3. Discussion

Detection of bloodstream infections by blood culture, identification of the causative organism, and subsequent measurement of antimicrobial susceptibilities is essential for the monitoring and treatment of bacteraemia. Our routine laboratory procedure requires initial blood culture, overnight incubation of 16–24 h on non-selective media [14] for bacterial identification, and a third culture step for AST, giving a total turnaround time of up to 53 h. Any measures that reduce testing time without compromising on detection and susceptibility results will be of benefit to patient management. We have found that AST results were not significantly affected by reducing the post-blood culture incubation period from up to 24 h to 6 h.

While 99.65% of results were shown to have the same susceptibility readings from 6 h cultures compared to 24 h cultures, there were three discrepant results. The first was an *Escherichia coli* specimen that yielded a ‘susceptible with increased exposure’ result from AST performed from 6 h cultures compared to a sensitive result from 24 h cultures. The AST plate produced from the 24 h culture was contaminated with *Bacillus simplex*, creating difficulty in zone measurement. Where contamination is observed, it is laboratory policy that these results are not reported, and the AST is repeated. This was plate contamination as opposed to the sample being polymicrobial. When being incorporated into routine laboratory work, a Gram-stained slide was set up as well as cultures. The plates were re-incubated following AST, and if there was any indication of polymicrobial infection from the Gram slide or the cultures, such as colonial morphology upon inspection, the proposed process would not be followed. The other two anomalous results were on the same *S. pneumoniae* Mueller Hinton Agar (MHA) plate and showed resistance to teicoplanin and vancomycin from shorter cultures compared to sensitivity from 24 h cultures. Isolates resistant to these antibiotics are considered as an unusual phenotype and are rare or not reported, and as such, this result was repeated and confirmed by an MIC-determining method [15]. All such anomalous results would have triggered reflex confirmatory checks. The discrepant results were not repeated, as despite being patient samples, they were tested retrospectively as part of the study and therefore did not require repeating as correct results were already reported beforehand.

Quantitative analysis of AST results showed more variation despite qualitative results matching from 6 and 24 h culture AST results. In the three isolates with the highest variability there was an 8 mm difference between zone sizes from 6 h and 24 h culture AST results. The isolates with the largest differences were proportionally sensitive isolates, with 87.5% of isolates with a difference of 4 mm–8 mm being sensitive. It should be noted this only represents 6.5% of all results, with 73% having a 0–1 mm difference, and 87.4% having a 0–2 mm difference. Regression analysis showed that sensitive/resistant strains or specific organism groups did not cause any significant differences to the results. Sensitive readings are more difficult to interpret on the AST plate, as zone edges are not as distinct.

The overall trend when considering all organisms and all antibiotics was very strong, as seen in Figure 1, where an R^2^ value ranging from 0.97 to 0.99 for each organism group indicates that antibiotic disc zone sizes from 6 and 6 h cultures closely correlated. When broken down into the 27 individual antibiotics used, the lowest R^2^ values of 0.58 for penicillin and 0.55 for linezolid may have been due to all organisms being sensitive to these antibiotics, and therefore, zone sizes were all large and clustered in one area of the scatter graph, reducing the likelihood of a strong correlation. Another explanation is natural variation in zone sizes. Control strains of *S. pneumoniae* ATCC 49619, *E. faecalis* ATCC 29212, and *S. aureus* ATCC 29213 had AST performed against a range of antibiotics daily as quality controls. Despite the same organism being tested daily, there was still a natural variation in zone sizes. Penicillin and linezolid all had a range of up to 6 mm, which zone sizes are expected to fall between.

The disc diffusion method is prone to variability. Zone sizes can be affected by multiple variables, including how far the antibiotic diffuses into the agar surrounding the disc. If a disc is not flat to the surface of the agar, or if the depth of the agar is varied, this can affect how far the antibiotic diffuses which will alter zone sizes. These zones are also affected by the molecular weight of the drug, as those that are lighter have higher diffusivity and are able to travel further through the agar, explaining why some antibiotics display larger zones [16]. Regression analysis showed that drug molecular weight did not make a significant difference to the results. Inoculant preparation and application can be prone to variation as the saturation of the swab is not controlled, and human error may occur in the streaking technique. Zone sizes were measured by a small number of staff members, and variation in reading technique is likely to occur, introducing marginal variation. Despite these factors, the interpreted susceptibility results did not seem to be affected. Nevertheless, given the mean 1.1 mm difference between the 6-h and overnight zone sizes, it would be advisable to initially trial the new method to ensure it does not impact on susceptibility results. Despite these variables, there are many benefits, including the use of cheap materials, the ability to perform all work within normal working hours, and a well-established method with accurate breakpoints that are updated yearly to mirror changes in resistance patterns. The ability to test up to six antibiotics on one plate benefits cost and workload while increasing the throughput.

Currently, in the Microbiology Laboratory in University Hospital Southampton, turnaround times for AST results vary. Any positive blood cultures that flag positive between 4 and 10 a.m. have a turnaround time of 53+ h. Any positive blood cultures that flag positive between 12 p.m. to 2 a.m. have a turnaround time of 45+ h. Figure 2 shows the stages and timing of each of these processes, which have been developed to meet incubation requirements and fit into laboratory working hours. The new method, with only 6 h post blood culture incubation, provides significantly reduced turnaround times. While this depends on the timing of the blood culture flagging positive to fit within working hours, minimum turnarounds of 26 h can be met, particularly benefitting those that flag positive before 9 a.m., as AST can be set up on the same day, saving up to 27 h.

There were some limitations to the study. This was a single-centre study, with the practical work predominantly carried out by two laboratory workers. This is, in some ways, applicable to real life, as more than one person would operate in the laboratory but requires training to ensure reproducible results. Performing triplicates of each sample and control strains would been useful to show reproducibility of the results; however, this was not possible due to resources and time constraints. Additionally, as previously mentioned, the disc diffusion method introduced variability in results. The discrepant results should have been repeated as it would be done in practice. It may have been beneficial to perform an alternative method to confirm results from both 6 h and 24 h cultures, such as the EUCAST broth microdilution method [17], to compare the Minimum Inhibitory Concentration (MIC). Despite this, the purpose of the study was to compare results to a validated EUCAST method with only a minor method change, which was successfully done. Control organisms were included that possessed enzymatic resistance methods (Table 2) to ensure that the same resistance methods were expressed when incubated for shorter times, which was successfully shown. Sensitive strains were not included, which would have been beneficial. The new process was based on blood cultures being processed at 9 a.m., due to the working hours in the laboratory where the study was performed. This allowed the most rapid turnaround time to be met. When introducing this process into practice, this will have to be adapted accordingly to the specific laboratory and may require additional out of hours work to keep turnaround times rapid. The process will require some additional training in laboratories that do not use procedures in line with EUCAST. Otherwise, training requirements are minimal. 

Improving the speed and accuracy of AST for blood cultures has focused on technological innovations to measure susceptibility either during or post-culture. Recently developed commercial systems have included measurement of bacterial cultures via laser nephelometry (Alfred60/AST, Alifax, Padua, Italy) [18], time-lapse microscopy (ASTar, Q-linea, Uppsala, Sweden) [19], morphokinetic cellular analysis (Accelerate Pheno, Accelerate Diagnostics, Tucson, AZ, USA) [20], and the detection of volatile organic compound release (Reveal-AST, Specific Technologies, Somerville, NJ, USA) [21]. These systems all give more rapid AST results than traditional susceptibility testing but require significant initial capital investment, additional consumables costs, and training of staff in the use of new methodologies.

In this study, we showed that turnaround times from a positive blood culture bottle to AST result could be significantly reduced by a simple method change that is cost-neutral and can be performed by existing staff without the need for additional training. When laboratories are under pressure to improve performance while working within strict budgets, modification of an existing method that saves up to 27 h testing time could be an attractive alternative to investing in a new automated AST system.

## 4. Materials and Methods

The study to improve AST turnaround times was carried out at the diagnostic microbiology laboratory at UHS NHS Foundation Trust, a busy tertiary hospital with >1200 beds.

### 4.1. Culture Plate Set Up and Organism Identification

At the start of the working day, all positive blood cultures from the previous 24 h were inoculated onto two Columbia Agar with chocolate horse blood (CHOC) and two Columbia Agar with horse blood (CBA) plates and incubated at 37 °C. To investigate the effect of reduced incubation time, one set of CHOC and CBA plates was removed after 6 h and used for MALDI ToF analysis (Bruker Biotyper, Bremen, Germany). The second set of plates received 24 h incubation before analysis. The bacterial identification was performed in accordance with supplier guidelines [22].

### 4.2. Performing AST According to EUCAST

Dependent on the organism identification, the relevant AST was performed by disc diffusion using EUCAST methodology, with MHA or Mueller Hinton and horse blood Fastidious agar (MHF). The EUCAST set-up was performed between 1 and 5 pm to ensure that the AST plates had incubation times of 16–20 h before being read the following morning at 9 a.m. AST plates were read according to EUCAST methodology and given a sensitive, resistant, susceptible with increased exposure, or ATU result, for each antibiotic [10]. On the same day, the second set of CHOC and CBA plates that had received 24 h incubation were removed from the incubator, the same AST procedure was performed, and the plates were read accordingly the following morning.

### 4.3. Data Collection

Based on MALDI-ToF identifications at 6 h, the study was continued until there were at least 20 comparisons for each of *Staphylococcus* spp. organisms (including *S. aureus*, *Staphylococcus epidermidis*, *Staphylococcus capitis*, *Staphylococcus haemolyticus*, and *Staphylococcus hominis*), Enterobacterales (including *Enterobacter cloacae*, *E. coli*, *Klebsiella aerogenes*, *K. pneumoniae*, *Proteus mirabilis*, and *Serratia marcescens*), and *Streptococcus/Enterococcus* spp. (including *E. faecalis*, *E. faecium*, *Streptococcus algalactiae*, *Streptococcus dysgalactiae*, *Streptococcus pyogenes*, and *S. pneumoniae*). Any sample cultures that were not identified at 6 h, or did not match one of the organism groups, were not included in the analysis. To investigate whether any trends in results varied among uncommon bacteraemic organisms, 20 blood culture bottles were inoculated with horse blood containing clinically significant isolates that caused pneumonia, such as *H. influenzae* and *M. catarrhalis*. These were loaded onto the BD Bactec FX blood culture machine and, once flagged positive, were treated in the same way as all other samples outlined above. Nine control strains, with specific resistance mechanisms as shown in Table 2, underwent the same procedure.

No results were communicated to the blood culture bench during the investigations to ensure results were not reported and did not interfere with patient samples and management.

### 4.4. Data Analysis

The results were collected and compared in two ways:Qualitative results of sensitive (S), resistant (R), susceptible with increased exposure (I), and ATU from AST of 6 and 24 h cultures, for each antibiotic tested on each specimen. Qualitative results were used to determine sensitivity, specificity, Positive Predictive Value (PPV), Negative Predictive Value (NPV), and accuracy of the 6-h culture method.Quantitative results of zone diameter sizes (mm) for each antibiotic for each specimen from 6 and 24 h cultures that had AST performed on them. R^2^ was calculated using Microsoft Excel software. Linear regression analysis was conducted using the lm function in R (version 4.3.2; R Core Team, Vienna, Austria).

## 5. Conclusions

The use of a shorter post-blood culture incubation period of 6 h in comparison to the conventional longer incubation before the preparation of AST provides a quicker turnaround time while still using calibrated and controlled methods for AST. Using representative organisms and a large sample size, this study confirmed the reliability of using a shorter culture time for AST. This simple process change could have a disproportionately positive impact on the care and treatment of patients with bacteraemia, resulting in a better prognosis for this often life-threatening condition and impacting more other areas of health care more widely, such as bed-occupancy and AMS.

## Figures and Tables

**Figure 1 antibiotics-13-01094-f001:**
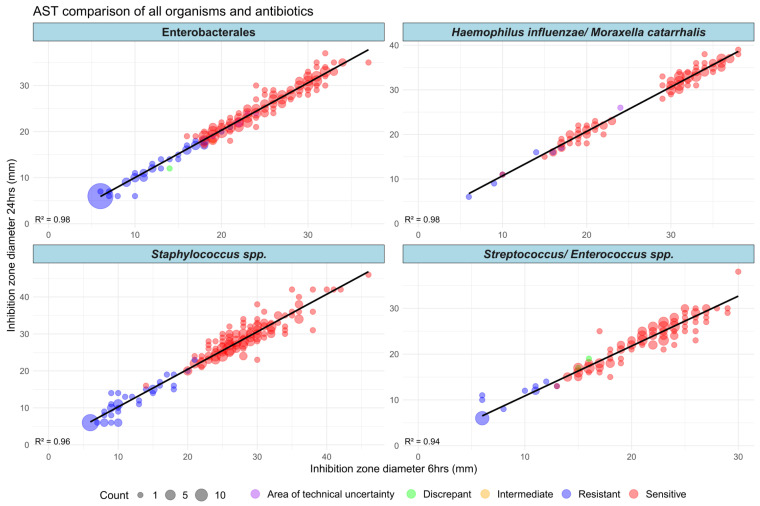
Comparison of AST inhibition zone sizes in mm for each organism group for all antibiotics from 6 h and 24 h culture plates. Individual points represent one organism result for one antibiotic. Where overlaid, this is indicated by the point size. A trendline and R^2^ value are included.

**Figure 2 antibiotics-13-01094-f002:**
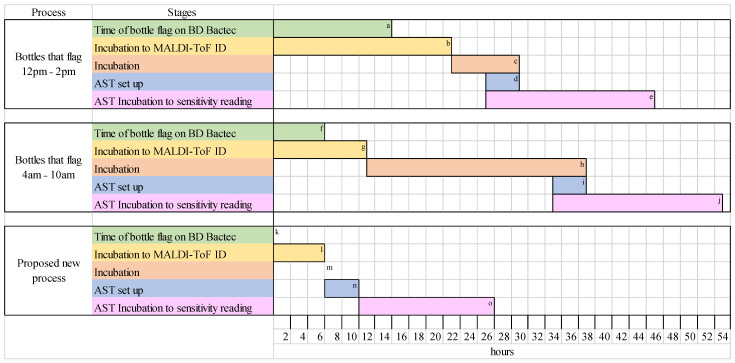
Current and proposed processes for positive blood cultures to AST results. ^a^ Positive blood cultures that flag between 12 p.m. to 2 a.m. ^b^ Minimum incubation time of 7 h, maximum incubation time of 21 h. ^c^ Minimum incubation time of 4 h, maximum incubation time of 8 h. ^d^ AST set up performed within a time range of 4 h. ^e^ Minimum incubation time of 16 h, maximum incubation time of 20 h. ^f^ Positive blood cultures that flag between 4 am to 10 am. ^g^ Minimum incubation time of 5 h, maximum incubation time of 11 h. ^h^ Minimum incubation time of 22 h, maximum incubation time of 26 h. ^i^ AST set up performed within a time range of 4 h. ^j^ Minimum incubation time of 16 h, maximum incubation time of 20 h. ^k^ The new process does not have a time range for the bottle to flag positive and is processed immediately. ^l^ incubation time of 6 h. ^m^ No incubation after MALDI ID as AST set up performed immediately. ^n^ AST set up performed within a time range of 4 h. ^o^ Minimum incubation time of 16 h, maximum incubation time of 20 h.

**Table 1 antibiotics-13-01094-t001:** Summary of qualitative results for 82 samples and nine control organisms.

		24-h AST Result			
		S	R	I	ATU
Six hour AST result	S	631		1 *	
	R	2 *	213		
	I			4	
	ATU				7

* Discrepant results.

**Table 2 antibiotics-13-01094-t002:** Control organisms and resistance mechanisms.

Organism	ATCC/NCTC Number	Resistance Mechanism
*S. aureus*	ATCC 33591	MRSA
*E. faecium*	NCTC 12202	VRE
*E. coli*	NCTC 13476	IMP
*K. pneumoniae*	ATCC 700603	ESBL
*K. pneumoniae*	NCTC 13439	VIM-1
*K. pneumoniae*	ATCC BAA-2814	KPC-3
*K. pneumoniae*	NCTC 13438	KPC-3
*K. pneumoniae*	NCTC 13442	OXA-48
*K. pneumoniae*	NCTC 13443	NDM-1

Abbreviations: Methicillin-resistant *Staphylococcus aureus* (MRSA), Vancomycin-resistant Enterococci (VRE), Imipenemase (IMP), Extended Spectrum Beta Lactamases (ESBL), Verona integron-encoded metallo-beta-lactamase 1 (VIM-1), Klebsiella pneumoniae carbapenemase (KPC), Oxacillinase 48 (OXA-48), New Delhi metallo-beta-lactamase 1 (NDM-1).

## Data Availability

Raw data are available through Appendix A. For confidentiality reasons, full data are available on request to the corresponding author and subject to ethical approval.

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
