# Peer review of "Improving Turnaround Times for Routine Antimicrobial Sensitivity Testing Following European Committee on Antimicrobial Susceptibility Testing Methodology in Patients with Bacteraemia"

_antibiotics, 2024, doi:10.3390/antibiotics13111094_

Round 1
Reviewer 1 Report
Comments and Suggestions for Authors
Comments for authors
This is my first-round review, and major issues should be clarified:
1. All affiliations are not completed.
2. Left “Antibiotic Susceptibility Test (AST)” only in the Abstract part; in the other part, you can use the abbreviated form. Check throughout the documents. Also, AMR, AMS, EUCAST, etc.
3. Please revise the writing style in the Materials and Methods section. It would be better to provide subsections 4.1, 4.2, … 4.4. To clarify the step and make the readers understand your work.
4. All figures are not clear.
Good Luck!
Reviewer 2 Report
Comments and Suggestions for Authors
Major points
- What are the advantages of this method compared to the already available RAST? EUCAST has already established clear guidelines for perfoming RAST directly from positive blood cultures. Compared to the method outlined here, RAST further shortens turnaround time as it is performed directly from positive blood cultures
This method: positive BC – 6 hr incubation for culture – setup 2 hr – 16-18 hr incubation for AST – total at least 26 hr
RAST: positive BC – 4-8 hr incubation for AST – total 4-8 hr
- A clear and extended comparison to RAST should be included in the manuscript including similarities, differences and advantages/ disadvantages for each method
- Line 22 – This is factually incorrect. EUCAST clearly specifies the incubation time for overnight cultures in the case of Disk Diffusion – “The inoculum suspension is prepared by selecting several morphologically similar colonies (when possible) from overnight growth (16–24 h of incubation)” available at https://www.eucast.org/fileadmin/src/media/PDFs/EUCAST_files/General_documents/Publications/Disk_diffusion_paper_printed_version_March_2014.pdf
- Line 81 – Some species are clearly underrepresented (e.g., S. capitis, E. cloacae etc). A table should be added highlighting the exact number of each bacterial species.
- Ideally, the AST results should have been confirmed through a MIC technique as a confirmation for both 6hr and 24hr culture methods (e.g., broth microdilution).
- Table 2 Control organisms – the addition of ATCC/NCTC resistant strains represents a strength of the manuscript. However, were the ATCC strains recommended by EUCAST for quality control purposes not tested?
- Methods – How was R2 calculated? There is no indication of the statistical software used
- Line 237 – add complete bacterial genus and species
Minor points
- author affiliation is missing
- Table 1 needs to be reformatted to improve readability
- Supplementary File – Ent 4 has no Vancomycin reading at 6 hr ?
Reviewer 3 Report
Comments and Suggestions for Authors
In the present study, the authors have applied the normal methodology of the European Committee on Antimicrobial Susceptibility Testing (EUCAST) disc diffusion AST method from positive blood cultures and compared the disc diffusion results from cultures with six hours incubation to those with incubation of 24 hours in the routine laboratory of a University Hospital in the UK. This is an interesting study, but there are some points that should be clarified before publication.
Major comments:
- The authors state that: ''EUCAST methodology outlines that disc diffusion should be performed on isolates from an overnight culture, without specifying incubation time in hours.'' (lines 21-23), ''The aim of this study is to improve and reduce turnaround times for AST results in positive blood cultures, without applying expensive technology, at a tertiary hospital.'' (lines 73-76) and ''The new method, with only 6-hour post blood culture incubation provides significantly reduced turnaround times'' (lines 178-179).
Nonetheless, EUCAST has developed a method for rapid AST (RAST), reading at 4, 6 or 8h and since April 2022 also after 16-20 hours incubation, directly from positive blood culture bottles, and a clinical trial in 55 laboratories was performed and published in 2020 [please at: https://www.eucast.org/rapid_ast_in_bloodcultures , Emma Jonasson, et al., Evaluation of prolonged incubation time of 16–20 h with the EUCAST rapid antimicrobial susceptibility disc diffusion testing method, Journal of Antimicrobial Chemotherapy, Volume 78, Issue 12, December 2023, Pages 2926–2932, https://doi.org/10.1093/jac/dkad332, ; Åkerlund A, Jonasson E, Matuschek E, Serrander L, Sundqvist M, Kahlmeter G; RAST Study Group. EUCAST rapid antimicrobial susceptibility testing (RAST) in blood cultures: validation in 55 European laboratories. J Antimicrob Chemother. 2020 Nov 1;75(11):3230-3238. doi: 10.1093/jac/dkaa333.)
Please revise these statements accordingly.
- The authors may also compare the RAST method used in this study with the reference method (broth microdilution method) for the control organisms, by calculating the errors (if any) on the total number of zones interpreted as S or R (%), such as: Minor error (mE; RAST=S or R and method=I); major error (ME; RAST= R and reference method=S); very major error (VME; RAST=S and reference method=R). Results in the ATU are exempted from error calculations since no categorization is performed [please see at: Åkerlund A, Jonasson E, Matuschek E, Serrander L, Sundqvist M, Kahlmeter G; RAST Study Group. EUCAST rapid antimicrobial susceptibility testing (RAST) in blood cultures: validation in 55 European laboratories. J Antimicrob Chemother. 2020 Nov 1;75(11):3230-3238. doi: 10.1093/jac/dkaa333.)
- The authors state that ''The larger variation did not seem to be caused by specific organism groups'' (lines 141-142) and ''...zone sizes are all large and cluster in one area of the scatter graph, reducing the likelihood of a strong correlation. Another explanation is natural variation in zone sizes (lines 148-149). The data presented in the summplemantary material may also be analysed by statistic methods (such as multivariate regression, clustering or principal component analysis) to explore diffent factors that may be associated with differences in the zone diameters, such as resistant v.s. susceptible strains, species characterization, drug molecular weight, and justify these statements.
- lines 126-127. The contamination was identified after 24hrs of incubation. Hence, it is possible to obtain false results by applying the RAST method (6hrs incubation) and in cases of polymicrobial samples. Moreover, direct MALDI-TOF MS has low performance with polymicrobial samples, compared with other molecular methods (please see at: Almuhayawi, M.S., Wong, A.Y.W., Kynning, M., Lüthje, P., Ullberg, M. and Özenci, V. (2021), Identification of microorganisms directly from blood culture bottles with polymicrobial growth: comparison of FilmArray and direct MALDI-TOF MS. APMIS, 129: 178-185. https://doi.org/10.1111/apm.13107). Gram-staining and microsopy to decide if the samples are mono- or polymicrobial would may be added as a step to decide to wheter or not proceed to RAST for obtaining more accurate results? Please clarify these issues.
- The authors state: ''The disc diffusion method is prone to variability'' (line 151). Experiments may be performed as triplicates to measure reproducibility of the results for the control strains.
Minor comments:
- Abstract. The background may be reduced and the aims of this study should be more specific. Additionally, please add the method used for organisms identification (MALDI ToF) in the Methods of the abstract.
- line 44: please correct as: ''Staphylococcus spp.'' in italics.
- line 46: please add the manufacturer for BD Bactec FX.
- lines 79-80: please correct and use italis as: ''Staphylococcus spp.'' , ''Enterococcus/Streptococcus spp.'', and write the species names in full on first occurence for ''Haemophilus influenzae'', ''Moraxella catarrhalis'' and other species throughout the text.
- line 113: please describe the ''improvement''
- line 119: please write the city/country for Bruker Biotyper).
- line 230: please write the version of the EUCAST breakpoint tables for AST interpretation.
- line 237: Do you mean the Influenza virus or ''''H. influenzae'' and ''Moraxella catarrhalis'' ?
- lines 124-126: please clarify if the discrepant results were not patient samples, as stated in line 187.
- The legends of the tables and figures do not need to be underlined.
Round 2
Reviewer 2 Report
Comments and Suggestions for Authors
While the manuscript has been visibly improved, especially in highlighting the advantages and disadvantages as compared to RAST, there are still some minor issues that need to be addressed.
- Figure 1 – replace antibiotic disc zone size with inhibition zone diameter
- I agree with the added paragraph comparing this method to RAST; however, one line requires rephrasing as it is factually incorrect “The disc diffusion plates also need to be read within five minutes of the incubation period” – if this refers to a reading immediately following the end of the incubation period, specify it as such
- Methods section 4.2 – Performing EUCAST makes no sense – replace with Performing AST according to EUCAST
- The comparison with RAST might be better suited for the introduction in order to present the different advantages and disadvantages of this novel method
Reviewer 3 Report
Comments and Suggestions for Authors
The authors have successfully replied to the comments and ammended the manuscript adequately for publication
Author Response
Thank you very much for the positive feedback.